# Lung Ultrasound for the Exclusion of Pneumothorax after Interventional Bronchoscopies—A Retrospective Study

**DOI:** 10.3390/jcm12041474

**Published:** 2023-02-12

**Authors:** Melanie Scarlett Mangold, Fabienne Rüber, Carolin Steinack, Fiorenza Gautschi, Jasmin Wani, Sascha Grimaldi, Daniel Peter Franzen

**Affiliations:** 1Department of Pulmonology, University Hospital Zurich, Raemistrasse 100, 8091 Zurich, Switzerland; 2Department of Internal Medicine, Spital Uster, Brunnenstrasse 42, 8610 Uster, Switzerland

**Keywords:** lung ultrasound, pneumothorax, interventional bronchoscopy, chest X-ray, diagnostic yield

## Abstract

A chest X-ray (CXR) is recommended after bronchoscopies with an increased risk of pneumothorax (PTX). However, concerns regarding radiation exposure, expenses and staff requirements exist. A lung ultrasound (LUS) is a promising alternative for the detection of PTX, though data are scarce. This study aims to investigate the diagnostic yield of LUS compared to CXR, to exclude PTX after bronchoscopies with increased risk. This retrospective single-centre study included transbronchial forceps biopsies, transbronchial lung cryobiopsies and endobronchial valve treatments. Post-interventional PTX screening consisted of immediate LUS and CXR within two hours. In total, 271 patients were included. Early PTX incidence was 3.3%. Sensitivity, specificity, and the positive and negative predictive values of LUS were 67.7% (95% CI 29.93–92.51%), 99.2% (95% CI 97.27–99.91%), 75.0% (95% CI 41.16–92.79%) and 98.9% (95% CI 97.18–99.54%), respectively. PTX detection by LUS enabled the immediate placement of two pleural drains along with the bronchoscopy. With CXR, three false-positives and one false-negative were observed; the latter evolved into a tension-PTX. LUS correctly diagnosed these cases. Despite low sensitivity, LUS enables early diagnosis of PTX, thus preventing treatment delays. We recommend immediate LUS, in addition to LUS or CXR after two to four hours and monitoring for signs and symptoms. Prospective studies with higher sample sizes are needed.

## 1. Introduction

Interventional bronchoscopy is essential for the diagnosis and treatment of various lung diseases. Some procedures bear an increased risk of pneumothorax (PTX). The reported incidence rate after transbronchial forceps biopsy (TBFB) is 2.9–7% [1,2,3,4], 5–20.2% after transbronchial lung cryobiopsy (TBLC) [5,6,7] and 14–39% after endobronchial valve (EBV) placement for bronchoscopic lung volume reduction (BLVR) [8]. Since PTX is a potentially life-threatening complication, the German and British guidelines recommend a chest X-ray (CXR) after TBFB if there is suspicion of PTX [9,10] and routine CXR after TBLC [9]. Most PTXs after EBV placement occur within 48 h [11], hence a hospitalisation of at least 72 h [11] and a CXR imaging within 4 h and, additionally, 24 h post-procedure [12] is advised. However, concerns regarding radiation exposure, expenses, staff requirements and infrastructure exist with CXR.

Lung ultrasound (LUS) is an upcoming technology increasingly used in emergency departments and intensive care units for PTX diagnosis [9] and has shown higher diagnostic accuracy than supine CXR [13]. The main advantage is its rapid availability at the bedside without having to move the patient, thus saving time in PTX diagnosis. The German S3-guidelines [9] suggest LUS as a possible alternative to CXR for the diagnosis of post-interventional PTX. However, this recommendation is based on studies that only included TBFBs, percutaneous transthoracic lung biopsies and thoracentesis [9]. To the best of our knowledge, there are currently no recommendations for the use of LUS for PTX diagnosis after TBLC or EBV treatment. Most lung centres primarily use routine CXR for the exclusion of post-interventional PTX [14]. The aim of this retrospective study was to investigate the diagnostic yield of immediate LUS compared to CXR for the exclusion of PTX after flexible bronchoscopies with TBFB, TBLC or EBV treatment.

## 2. Materials and Methods

### 2.1. Overall Study Design and Patient Selection

This retrospective single-arm study took place at the University Hospital of Zurich, a tertiary care centre in Switzerland. Prior to this study, an institutional standard operating procedure (SOP) was established at the University Hospital of Zurich where, as of 1 September 2019, after TBFB, TBLC or EBV treatment every patient receives immediate LUS and routine CXR within two hours for clinical evaluation of post-procedural PTX; previous to this SOP, patients only received routine CXR. Patient- and procedure-related data, as well as CXR and LUS findings, were retrospectively extracted from institutional patient record files. The Ethics Committee’s approval was waived as the project did not fall within the scope of the Human Research Act (BASEC-Nr. Req-2020-00559). All patients included in this study gave their consent by written general consent.

Patients undergoing scheduled flexible bronchoscopies with TBFB, TBLC or EBV treatment between 1 September 2019 and 30 September 2020 were enrolled in this study. Exclusion criteria for this study were (1) patients under 18 years of age, (2) no general consent given or (3) change of planned intervention without undergoing TBFB, TBLC or EBV treatment.

### 2.2. Procedure and Screening for Pneumothorax

All procedures were performed with a flexible bronchoscope (BF-1TH190, BF-UC180F, Olympus Medical Systems, Tokyo, Japan) under propofol sedation or general anaesthesia. TBFBs were obtained with mini or cup biopsy forceps (FB-433D, FB-231D, Olympus Medical Systems, Tokyo, Japan). A cytology brush (BC-202D-2010, Olympus Medical Systems, Tokyo, Japan) was used for bronchial brushing. TBLCs were obtained with a 1.7 mm or 2.4 mm diameter cryoprobe (Erbecryo 2, Erbe Elektromedizin GmbH, Tuebingen, Germany). For BLVR treatment, Zephyr EBVs (4.0 EBV, 4.0-LP EBV, 5.5 EBV, 5.5-LP EBV, PulmonX Inc., Redwood City, CA, USA) were used. All TBFBs and TBLCs were used under fluoroscopic guidance (Oec One, GE Healthcare, Chicago, IL, USA). After the procedure, all patients were screened for PTX with LUS, while still under sedation or general anaesthesia in the operating room. From there, patients were taken to the recovery room for clinical monitoring of oxygen saturation, pulse, blood pressure, acute chest pain or dyspnoea. Within two hours post-procedure all patients received a routine CXR image in either inspiratory upright posteroanterior and lateral view or in supine anteroposterior view, depending on the patient’s ability to stand upright. Inpatients were then brought to the ward and outpatients were clinically monitored and discharged after PTX had been ruled out by imaging. All patients were informed about the risks of delayed PTX.

The LUS was performed by the bronchoscopist on the patient in the supine body position. The transducer was placed longitudinally at the highest point of the anterior thorax, then moved along the intercostal spaces of the parasternal line and laterally, between the anterior and posterior axillary line, checking for lung sliding and lung point in 2D B-mode, and for seashore or stratosphere sign in M-mode. The hemithorax without intervention was examined first and then compared to the hemithorax with intervention. An advanced ultrasound machine (LogiQ P9 R3, GE Healthcare, Chicago, IL, USA) and a 10 MHz linear probe were used. Nine bronchoscopists, four residents and five senior physicians, were involved in this study. LUS was assessed by the senior physician or by both, the resident and the senior physician, to reliably detect or rule out PTX.

The LUS artefacts used for PTX detection were loss of lung sliding with the presence of lung point in B-mode, and stratosphere sign in M-mode (Figure 1) [15]. The lung point was used to estimate PTX size with LUS; the further laterally the lung point sign was detected on the thorax, the bigger the PTX [16]. For PTX exclusion, the presence of lung sliding in B-mode with a seashore sign in M-mode was used [15].

A pleural drain was indicated in the case of large, progressive or persistently symptomatic PTXs and was inserted with the aid of bedside LUS. Small PTXs were usually treated conservatively with oxygen and analgesics. Immediate placement of pleural drain after bronchoscopy, while the patient was still sedated or anaesthetised in the operating room, was only performed if a definite diagnosis of PTX was possible with LUS or in case of clinical instability or impaired gas exchange. In stable patients with inconclusive PTX on LUS, an immediate CXR was taken and the appropriate further procedure was evaluated. A threshold of two centimetres interpleural distance, measured on CXR at the level of the lung hilum was used to distinguish between small and large PTXs, according to the British Thoracic Society [17].

### 2.3. Reference Standard

The reference standards used for this study were CXR findings, which indicated the presence or absence of PTX irrespective of LUS finding. However, in case of inconclusive initial CXR findings or discrepancies between symptoms and imaging findings, a follow-up CXR was ordered and the final diagnosis of PTX was made, as summarised in Table 1.

Symptoms suggestive of PTX were defined as new onset or worsening of chest pain or dyspnoea. Patients with EBV placement received a follow-up CXR in upright PA and lateral view 24 h post-procedure. If PTX was present on either immediate LUS or routine CXR within two hours, it was considered an early PTX. In contrast, if a follow-up CXR detected a PTX that was not present on either immediate LUS or routine CXR within two hours, this was termed a delayed PTX [18].

All physicians performing immediate LUS were blinded to the reference standard, as CXR was performed afterwards. The CXR images were interpreted by routine radiologists, blinded to the LUS findings.

### 2.4. Statistical Analysis

SPSS Statistics for Windows 27.0 (IBM, Armonk, NY, USA) was used for statistical analysis. Data are presented as counts, percentages or mean with standard deviation (SD), as appropriate. The confidence interval (CI) was set to 95%, and *p*-values less than 0.05 were considered significant. *p*-Values were calculated using Pearson’s chi-square test for categorical variables, Fisher’s exact test for binary variables and independent sample *t*-test for continuous data.

## 3. Results

From September 2019 to September 2020, there were 299 eligible patients undergoing TBFBs, TBLCs or EBV placement. From this number, 28 patients were excluded; 20 patients had no written documentation of LUS findings, 2 patients refused CXR imaging, 1 patient had neither LUS nor CXR imaging, 1 patient received routine CXR after more than 2 hours post-procedure, and 4 patients did not have an interventional bronchoscopic procedure as originally planned. The study flowchart is shown in Figure 2.

In total, 271 patients (mean age 62 years, 43% female) were included, the patient characteristics are shown in Table 2. The most common indication for flexible bronchoscopy was suspected pulmonary malignancy (40%) and the following numbers of procedures were performed: 164 TBFBs, 39 TBLCs, 32 TBFBs combined with TBLCs and 36 EBV treatments. The procedural data are shown in Table 3.

Immediate post-procedural LUS detected 8 patients with LUS artefacts consistent with PTX and 263 patients where PTX was ruled out. The reference standard detected 9 early PTXs (3.3%) and ruled out PTX in 262 patients (96.7%). The reference standard is the result of one, or if necessary multiple CXR images, which were performed according to Table 1. Out of the nine early PTXs, seven needed pleural drain (2.9%) (1 TBFB, 2 TBFB + TBLC, 3 TBLC, 1 EBV), of which two patients were symptomatic. The remaining two PTXs (2 TBFB) were treated conservatively.

Initial CXR imaging was taken within two hours of bronchoscopy, after a mean time of 74 min (SD 59 min). In 166 patients (61.3%), the imaging was performed in the erect posteroanterior and lateral view, while in 105 patients (38.7%) in the supine anteroposterior view. After the initial CXR imaging, some patients required a repeat CXR according to Table 1. Additionally, five delayed PTXs (1.8%) were detected. These five delayed PTXs were absent on both, the initial LUS and CXR imaging, and were detected between 20 h and more than 8 days post-procedure. One delayed PTX was asymptomatic and treated conservatively (1 EBV). Four delayed PTXs were symptomatic (2 EBV, 1 TBFB, 1 TBLC); two of them needed pleural drains, and the remaining two were treated conservatively.

In total, 17 patients (6.3%) showed symptoms suggestive of PTX, either chest pain or dyspnoea, or both; these were 2 early PTXs (2 TBFB + TBLC), 4 delayed PTXs (1 TBFB, 1 TBLC, 2 EBV) and 11 patients (3 TBFB, 1 TBLC, 7 EBV) where PTX was ruled out by repeat CXR, according to Table 1. Compared to the reference standard for early PTX diagnosis, the diagnostic accuracy of LUS was as follows: sensitivity 66.67% (95% CI 29.93–92.51%), specificity 99.24% (95% CI 97.27–99.91%), positive predictive value (PPV) 75.0% (95% CI 41.16–92.79%), negative predictive value (NPV) 98.86% (95% CI 97.18–99.54%), positive likelihood ratio 87.33 (95% CI 20.37–374.50) and negative likelihood ratio 0.34 (95% CI 0.13–0.85). Table 4 shows the 2 × 2 cross-tabulation of LUS compared to the reference standard.

With LUS imaging there were two false-positive and three false-negative incidents. Out of those three false-negative cases, one patient was symptomatic, though all three patients received a pleural drain. Interestingly, one of these patients received a second LUS imaging before placing the pleural drain, more than 6 h post-procedure, where LUS was able to detect the PTX. The other two false-negative cases requiring pleural drain did not have a second LUS imaging.

Surprisingly, there was one false-negative PTX with initial CXR in posteroanterior and lateral view, which was correctly diagnosed as PTX with immediate LUS. This patient developed chest pain and dyspnoea 20 h post-procedure. The follow-up CXR then detected a tension-PTX which needed a pleural drain. Furthermore, there were three false-positive cases with initial CXR. These three false positives were all taken in the supine anteroposterior view. The radiologist suspected PTX but was inconclusive, immediate LUS had ruled out PTX though. With follow-up CXR in the erect posteroanterior and lateral view, PTX was ruled out.

In two patients, the physician suspected a PTX during bronchoscopy using fluoroscopy. The immediate LUS was able to confirm the PTX. These two patients received pleural drains in the operating room while still under general anaesthesia or sedation.

## 4. Discussion

LUS is increasingly used in emergency departments and intensive care units for the diagnosis or exclusion of PTX [9]. However, there is a lack of data concerning the use of LUS after flexible bronchoscopies. In the present study, both CXR and LUS were performed serially in patients after interventional bronchoscopies for 13 months at a single tertiary care centre. The total PTX incidence (5.2%, 14/271), consisting of nine early PTXs and five delayed PTXs, was comparable to others [14]. Furthermore, the overall observed PTX rate after TBFB (2.4%, 4/164; three early PTXs, one delayed PTX) and TBLC (10.3%, 4/39; three early PTXs, one delayed PTX) was similar to previous studies [1,2,3,4,5,6,7], though the PTX rate after EBV placement (11.1%, 4/36; one early PTX, three delayed PTXs) was comparably lower [8]. The PTX rate after combined procedures of TBFB and TBLC was 6.3% (2/36; two early PTXs).

Recent studies have shown high diagnostic accuracy for post-procedural LUS; immediate LUS after TBFB reached a sensitivity and specificity of 100% [4], while LUS two hours after TBFB had a sensitivity of 75–100% and a specificity of 83–93% [3,19]. One study [20] described a sensitivity of 21.1% and a specificity of 99.2% for immediate LUS after TBLC, while another study [21] had a sensitivity of 90% and a specificity of 94% for LUS 3 h after TBLC.

The different timings of LUS in these studies show inconclusive results on whether earlier or later LUS is more accurate for the diagnosis of post-procedural PTX. However, the exact timing of LUS could have a significant impact on its diagnostic accuracy. To our knowledge, there are currently no guideline recommendations for the optimal timing of LUS following bronchoscopic procedures for PTX diagnosis. CXR imaging after bronchoscopies is recommended no earlier than one hour post-procedure [22] and optimally within four hours [9], while the BTS Guidelines stated that 40% of PTXs may be delayed up to two hours [10]. The question, therefore, arises if LUS should be performed after two to four hours, similarly to CXR, to increase its diagnostic accuracy. A recent study by Eisenmann et al. [14] performed LUS immediately, as well as 2 h post-procedure in 115 patients with TBFBs, TBLCs, EBV or coil treatments. The early PTX rate was 3.47%, and LUS had a sensitivity of 75% and a specificity of 100%. Interestingly, the authors found no difference in the diagnostic accuracy of LUS when performed immediately or after 2 h. However, to recommend the optimal timing for LUS in clinical practice, further studies with a larger sample size comparing immediate to subsequent post-procedural LUS need to be conducted.

Eisenmann’s study design [14] is comparable to ours; however, they had higher diagnostic accuracy for immediate LUS. A reason for this might be interobserver variability, as they had one experienced physician, while our study had nine different physicians perform the LUS. Our diagnostic accuracy could have been impacted by the varying experience of LUS in patients who may have some pleural deformities due to their underlying disease. The difference in expertise represents the situation in most clinics, it is, therefore, important to ensure that all physicians are well trained in the use of LUS before introducing it for post-interventional PTX diagnosis. One should also be aware that there can be misjudgement in the beginning, as there is a learning curve with LUS.

Despite the low sensitivity, immediate LUS had a clinical impact on two patients that benefited from immediate pleural drain insertion while still sedated or anaesthetised in the operating room. Pleural drain insertion is known to be very painful for most patients [23]. Another patient had PTX diagnosed by immediate LUS, which the initial CXR did not detect; a follow-up CXR later detected a tension-PTX, thus delaying treatment.

These cases show that immediate LUS after bronchoscopy could benefit the patient in terms of safety, comfort and avoiding treatment delay of PTX. Another advantage of immediate LUS is its low cost and quick bedside availability. A recent study [20] found similar results for immediate LUS following TBLC. While the authors also noted low sensitivity for LUS, they described the immediate treatment option and its clinical implications as the main benefit of immediate LUS.

Today, many lung centres still use routine CXR to rule out PTX after bronchoscopies with increased risk. Some studies [1,2] have shown that routine CXR after TBFB may not be necessary for asymptomatic patients. Furthermore, the German [9] and British [10] guidelines do not recommend routine CXR after TBFB, but instead, recommend CXR imaging only in symptomatic patients. In our data, five out of seven patients requiring pleural drain for treatment of early PTX showed no clinical symptoms suggesting PTX. However, oxygen desaturation was not taken into consideration in our analysis which Izbicki et al. [1] described in their study as statistically significantly associated with PTX, together with acute or worsening chest pain. According to the guideline recommendations [9,10], we could envision a combined approach of LUS along with clinical signs and symptoms to replace routine CXR in patients after TBFB in the future.

Besides the above-mentioned flaws and the retrospective study design, there are further limitations to consider. Particularly, the number of events (PTXs) was low. Thus, the sample size was too small to draw robust conclusions. Another limitation was that not all CXR images were acquired in the same view and therefore CXR images in supine anteroposterior view may have reduced the diagnostic accuracy of the initial CXR, as erect CXR imaging in posteroanterior and lateral view is superior for diagnosis or exclusion of PTX [9,24]. This is an important observation since frail patients after bronchoscopic procedures and patients admitted to intensive care units or intermediate care may initially receive a supine anteroposterior CXR to rule out PTX. The reduced diagnostic accuracy of anteroposterior CXR could delay appropriate treatment and result in cumulative higher radiation exposure that could have been prevented by CXR imaging in posteroanterior and lateral views. Another limitation was the rare use of CT. Recent studies used CT scans to show that CXR misses some post-interventional PTXs which LUS detects [4,25,26,27]. In our study, a chest CT was carried out only in very rare cases, since it was not indicated within the given clinical context. For a future prospective study, we recommend CT imaging when there is a mismatch between CXR and LUS findings, although the higher radiation exposure must be considered.

## 5. Conclusions

Despite its low sensitivity, immediate LUS allows for early diagnosis of PTX after TBFBs, TBLCs and EBV treatments while still in the operating room, thus preventing treatment delays. We recommend performing an immediate LUS, followed by either LUS or CXR in posteroanterior and lateral view, two to four hours after TBFBs, TBLCs or EBV treatments, and monitoring for clinical signs and symptoms suggestive of PTX. A prospective study with a higher sample size and further research on the ideal time for LUS imaging after bronchoscopies is needed.

## Figures and Tables

**Figure 1 jcm-12-01474-f001:**
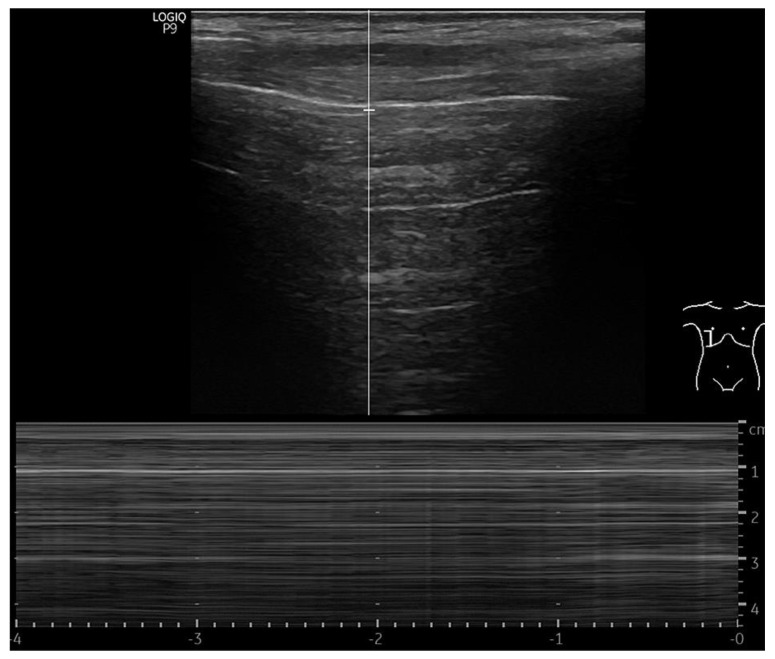
Stratosphere sign in M−mode caused by absence of lung sliding in pleural line during breathing, suggesting pneumothorax.

**Figure 2 jcm-12-01474-f002:**
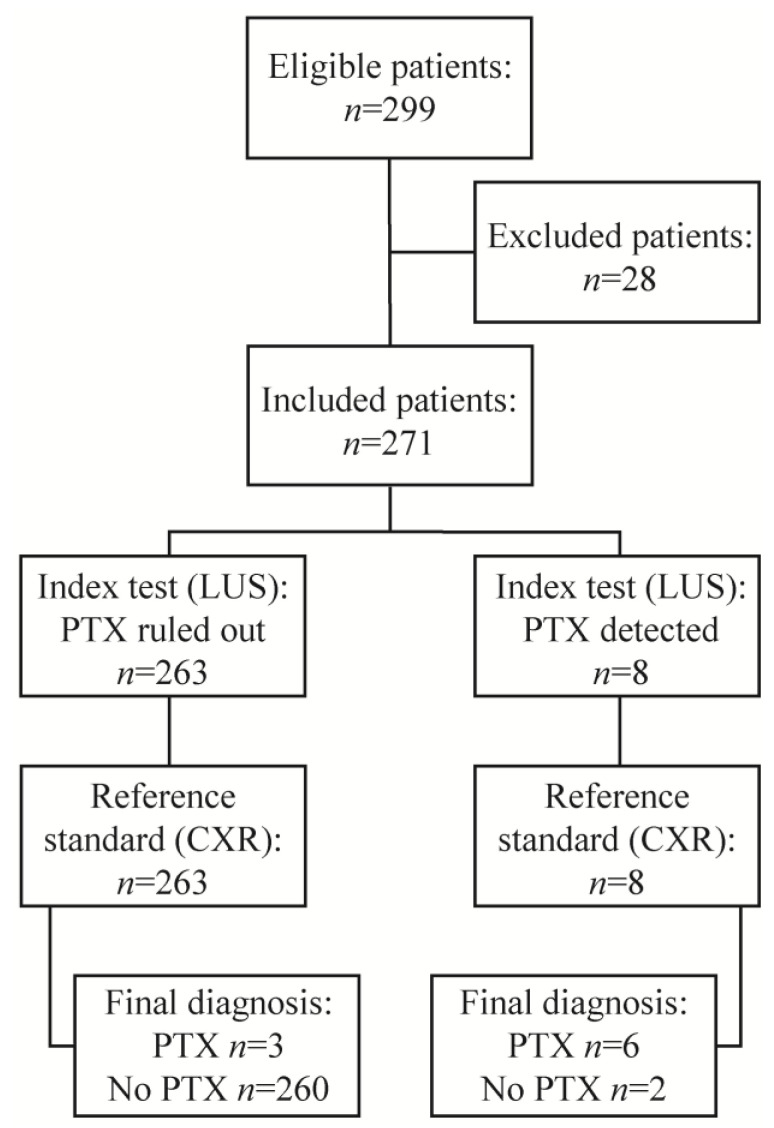
Study flowchart. LUS, lung ultrasound; PTX, pneumothorax; CXR, chest X-ray.

**Table 1 jcm-12-01474-t001:** Final diagnosis of pneumothorax (reference standard).

Final Diagnosis *	Initial CXR	Signs/Symptoms	Repeat CXR
No PTX	Negative	Negative	N/A
No PTX	Negative	Positive	Negative
No PTX	Positive	Negative	Negative
No PTX	Inconclusive	Positive/negative	Negative
PTX	Negative	Positive	Positive
PTX	Positive	Positive	N/A
PTX	Inconclusive	Positive/negative	Positive

* Reference standard for the purpose of the study. CXR, chest X-ray; PTX, pneumothorax; negative, absent CXR findings or signs/symptoms of PTX; positive, positive CXR findings or signs/symptoms of PTX; N/A, not applicable.

**Table 2 jcm-12-01474-t002:** Baseline characteristics.

Variables	Total (*n* = 271)	Early PTX (*n* = 9)	*p*-Value
Age (years)	62 (14)	60.2 (10.0)	0.773
Sex			0.006 *
Male	155 (57.2)	1 (11.1)	
Female	116 (42.8)	8 (88.9)	
BMI (kg/m^2^)	25.0 (4.8)	22.8 (3.6)	0.145
Smoking status			0.153
Never-smoker	83 (30.6)	2 (22.2)	
Former smoker	132 (48.7)	7 (77.8)	
Current smoker	56 (20.7)	0 (0)	
Indication for flexible bronchoscopy			0.254
Suspected pulmonary malignancy	109 (40.2)	2 (22.2)	
Surveillance after lung transplant	29 (10.7)	1 (11.1)	
Lung volume reduction	36 (13.3)	1 (11.1)	
Suspected ILD other than sarcoidosis	30 (11.1)	3 (33.3)	
Suspected sarcoidosis	26 (9.6)	0 (0)	
Suspected infectious disease	22 (8.1)	1 (11.1)	
Suspected non-infectious inflammatory disease	12 (4.4)	0 (0)	
Other indications	7 (2.6)	1 (11.1)	

Data are displayed as *n* (%) or mean (SD), respectively. PTX, pneumothorax; ILD, interstitial lung disease. * *p* < 0.05.

**Table 3 jcm-12-01474-t003:** Procedural data.

Variables	Total (*n* = 271)	Early PTX (*n* = 9)	*p*-Value
Type of intervention			0.226
TBFB	164 (60.5)	3 (33.3)	
TBLC	39 (14.4)	3 (33.3)	
TBFB + TBLC	32 (11.8)	2 (22.2)	
EBV	36 (13.3)	1 (11.1)	
Bronchial brushing	82 (30.3)	2 (22.2)	0.728
Anaesthesia			0.745
Sedation	163 (60.1)	5 (55.6)	
General anaesthesia	108 (39.9)	4 (44.4)	
Experience level of operator			1.00
Senior	257 (94.8)	9 (100.0)	
Resident	14 (5.2)	0 (0)	

Data are displayed as *n* (%). PTX, pneumothorax; TBFB, transbronchial forceps biopsy; TBLC, transbronchial lung cryobiopsy; EBV, endobronchial valve treatment. Bronchial brushing was additional to some TBLB/TBLC procedures.

**Table 4 jcm-12-01474-t004:** Post-procedural lung ultrasound compared to the reference standard: 2 × 2 table.

Lung Ultrasound	Reference Standard		Total
	Pneumothorax +	Pneumothorax −	
Pneumothorax +	6	2	8
Pneumothorax -	3	260	263
Total	9	262	271

Pneumothorax +, pneumothorax detected; Pneumothorax −, pneumothorax ruled out.

## Data Availability

The data presented in this study are openly available upon reasonable request.

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
