# Peer review of "Lung Ultrasound for the Exclusion of Pneumothorax after Interventional Bronchoscopies—A Retrospective Study"

_jcm, 2023, doi:10.3390/jcm12041474_

Round 1

Reviewer 1 Report

Dear Editor and Authors,

Thank you for the opportunity to review this manuscript.

LUS has become an increasingly used tool for the diagnosis of pneumothorax, especially in patients in whom the chest X-ray in the supine position has a lower sensitivity. For this reason, I believe that the authors address a very current topic, the immediate use of LUS in the exclusion of pneumothorax after interventional bronchoscopies. From a clinical point of view, the early diagnosis of the pneumothorax is very important both in case of a massive pneumothorax when the chest tube placement can be performed while the patient is still anesthetized, as well as in case of a minor PTX, which can be missed on the chest X-ray, thus allowing careful monitoring of the patient and subsequent LUS/CXR in order to establish the therapeutic management. Even if the study is a retrospective one, and the number of patients with PTX is low, each report can bring a benefit in the wide use of postprocedural LUS.

However, there are some minor concerns that the authors should adress in order to improve the manuscript:

1.       At Materials and methods, at point 2.1. paragraph 2, the authors wrote the inclusion criteria for TBLCs. Considering that the study was not performed only on patients who underwent TBLC and there are no data regarding patients who qualified for TBLC or were excluded based on those criteria, the authors should reconsider their relevance for the purpose of this study. Their presence before the exclusion criteria from the study can be confusing for the readers.

2.       At 2.3. Reference Standard line 130, The authors stated that "CXR images were interpreted by routine radiologists, blinded to the LUS findings". Given that this was a retrospective study, how were they sure that the doctors who performed LUS did not consult the radiologists when performing the CXR? Especially in patients in whom LUS reveals a pneumothorax or is inconclusive, a communication with the radiologist is usually useful in medical practice. If the authors cannot say this fact with certainty, they better remove the statement "blinded to the LUS findings".

3.       In Figure 2 (Study flowchart), I think that the Authors reversed the number of PTX ruled out (8) with those detected (263).

Author Response

Thank you very much for your review. Please see the attachment for our point-by-point response.

Reviewer 2 Report

In this retrospective study, Mangold et al have addressed a relevant topic with the aim of evaluating to what extent transthoracic ultrasound can be used to exclude pneumothorax after interventional bronchoscopy. This study is particularly relevant to evaluate the diagnostic value of a readily available, inexpensive procedure without radiation exposure.

1. Introduction:

The introduction clearly directs to the topic and presents the data basis in a structured manner.

2. Material and Methods: 

The authors describe that a standard operating procedure (SOP) for the use of LUS after interventional bronchoscopy was defined before initiating the study. Also, written informed consent was obtained from all patients. This suggests a principally prospective study approach. In contrast, the data collected on LUS, CXR, patient- and procedure-associated data were only evaluated retrospectively.

Perhaps the authors can comment on why the approach, which seems prospective in principle, was not also pursued prospectively in the sense of a quality assurance measure, but the data were only evaluated retrospectively.

The authors describe the LUS immediately following bronchoscopy, while the patient was still under ongoing sedation/general anesthesia. LUS is undoubtedly an examination that can be performed quickly in most cases, and the rationale is to insert the pleural drainage immediately in the event of a detected pneumothorax under continued sedation. Nevertheless, this procedure represents a prolongation of anesthesia that would not actually be necessary. Perhaps the authors can comment on why the LUS was not performed immediately after the patient awoke while still in the examination room. This would also allow clinical parameters used for the decision for/against drainage to be evaluated in awake and spontaneously breathing patients.

Statistical analysis: the authors describe the software used for statistical analysis. Please be precise about which tests were used (especially when comparing significance of multiple variables, such as in Table 2, 3).

Table 1. N/A - name abbreviation in legend.

3. results

Figure 2: 

The description under LUS is mixed up:

Index test (LUS): PTX detected n=263 - is not "ruled out" meant here?

Index test (LUS): PTX ruled out n=8 - is not "detected" meant here?

Table 2: Legend: TBLB, transbronchial lung biopsy does not appear in the table.

The focus of the presentation of results is clearly placed on the primary detection of early PTX. However, the discussion correctly also considers the development of delayed PTX. This leads to different reference values, for example regarding the incidence of PTX. Possibly, an additional presentation of the 5 cases with delayed PTX would provide a clearer overview here.

193 - "The other two false-positive cases requiring pleural ..." - reference is made here to the three false-negative cases, so this should be corrected.

198-202 It is problematic to report the cases reported here in supine antero-posterior view as false positives, as this imaging technique is suboptimal for detecting/excluding PTX, as also presented in the Discussion. The presentation here implies that the LUS "correctly" did NOT detect PTX and the x-ray incorrectly detected PTX. The supplemental x-ray in adequate position and thus optimal illumination, erect posterior and lateral view, showed no PTX as did the LUS. Thus, the CRX should not be declared as generally false positive in these cases. 

4. discussion

211 - The total PTX incidence (5.2%) .... - the information is correct with reference to the early and delayed PTX according to the Reference Standard with 9+5=16/271 = 5.2%. However, these figures are "difficult" to extract from the results section, since the delayed PTX do not appear in the tables. Thus, the reader must first research how this incidence figure comes about. A clearer presentation would be helpful here.

212 - The same applies to the incidence figures of PTX after TBFB and TBLC as well as after EBV. Here, the reader is forced to pick out according to (early or delayed) which incidences in relation to which number of examinations led to the percentages. Again, a clearer presentation is requested.

 261 - In the discussion the clinical symptoms of individual patients with PTX are given, here a presentation of the clinical symptoms of all patients (with and without PTX) would be desirable.

From 267 - Clear naming of the limitations of the study - limited number of events, different, partly suboptimal technique of CXR. This is fine.

Author Response

(The authors gave the same response as above.)
